# DISENTANGLEMENT AND GENERALIZATION UNDER CORRELATION SHIFTS

**Christina M. Funke**[*]
University of Tübingen

**Paul Vicol**[*]
University of Toronto
Vector Institute

**Kuan-Chieh Wang**
University of Toronto
Vector Institute

**Matthias Kümmerer**[†]
University of Tübingen

**Richard Zemel**[†]
University of Toronto
Vector Institute

**Matthias Bethge**[†]
University of Tübingen

## ABSTRACT

Correlations between factors of variation are prevalent in real-world data. However, often such correlations are not robust (e.g., they may change between domains, datasets, or applications) and we wish to avoid exploiting them. Disentanglement methods aim to learn representations which capture different factors of variation in latent subspaces. A common approach involves minimizing the mutual information between latent subspaces, such that each encodes a single underlying attribute. However, this fails when attributes are correlated. We solve this problem by enforcing independence between subspaces conditioned on the available attributes, which allows us to remove only dependencies that are not due to the correlation structure present in the training data. We achieve this via an adversarial approach to minimize the conditional mutual information (CMI) between subspaces with respect to categorical variables. We first show theoretically that CMI minimization is a good objective for robust disentanglement on linear problems with Gaussian data. We then apply our method on real-world datasets based on MNIST and CelebA, and show that it yields models that are disentangled and robust under correlation shift, including in weakly supervised settings.

## 1 INTRODUCTION

Most research on disentanglement has assumed that the underlying factors of variation in the data are *independent* (e.g., that factors are not correlated). However, this assumption is often violated in real-world settings: for example, in domain adaptation, the class distribution often shifts between domains (yielding a correlation between the class and domain); in natural images, there is often a strong correlation between the foreground and background (Beery et al., 2018), or between multiple foreground objects that tend to co-occur (e.g., a keyboard and monitor) (Tsipras et al., 2020; Beyer et al., 2020). Importantly, correlated data occurs in areas that affect people's lives, including in healthcare (Chartsias et al., 2018) and fairness applications (Madras et al., 2018; Creager et al., 2019; Locatello et al., 2019a), and correlation shifts in these applications are common (e.g., demographics are likely to differ from one hospital to another).

The goal of disentanglement is to encode data into independent subspaces that preferably match the ground truth generative factors. A common approach (used in ICA, PCA, and VAEs) is to ensure that the latent subspaces share as little information as possible, for example by minimizing the mutual information (MI) between them. However, recently it has been shown that this fails to disentangle correlated factors (Träuble et al., 2020). Several works have sought to address this by introducing partial supervision (Träuble et al., 2020; Shu et al., 2019; Locatello et al., 2020b). Here, we show that even under *full* supervision, minimizing the MI can fail: it is impossible to encode generative factors into independent subspaces if they are correlated in the training data. To address this, we propose minimizing the MI between subspaces *conditioned* on the correlated attributes. The goal of this work is to identify and explain the behaviors of different objective functions for correlated and noisy data in a systematic fashion.

---

[*]Equal contribution. [†]Shared senior authors.

| | Base | Base + MI | Base + CMI |
|---|---|---|---|
| **Variance Explained, Training (Corr = 0.8)** | 91.9% | 69.8% | 90.9% |
| **Variance Explained, Test (Corr = 0)** | 87.6% | 65.0% | 90.9% |

Table 1: Linear regression under correlation shift for each of the objectives *Base*, *Base+MI*, and *Base+CMI*. Here, the observations and predictions are 2D. Performance under correlation shift drops for optimal regression. The optimal solution under the constraint of minimal MI fails to model the in-distribution correlated training data. The solution with minimal *conditional* MI maintains consistent performance under correlation shift.

## 2    BACKGROUND & RELATED WORK

Disentangled representation learning is often studied in the unsupervised setting, where the ground-truth factors of variation are unknown. Widely-used approaches for this include variational autoencoders (VAEs) (Kingma & Welling, 2013) and their variants (beta-VAE (Higgins et al., 2017), TC-beta-VAE (Chen et al., 2018), FactorVAE (Kim & Mnih, 2018), etc.). However, it was shown by Locatello et al. (2019b) that *purely unsupervised* disentanglement may not be possible. In this paper, we focus on comparing MI and CMI minimization in the fully-supervised setting.

The goal of domain adaptation and generalisation is closely related as it attempts to learn representations from multiple source domains that transfer to known (e.g., adaptation) or previously unseen (e.g., generalization) target domains. This is done by either learning domain-invariant representations which discard domain information (Tzeng et al., 2017) or by learning disentangled representations, with latent subspaces that correspond to the domain and the class, respectively (Peng et al., 2019; Ilse et al., 2020; Liu et al., 2018). For the latter approach, disentanglement is achieved by minimizing the mutual information between latent subspaces (Cheng et al., 2020; Gholami et al., 2020; Nemeth, 2020). Additional related work is discussed in Appendix A.

## 3    DISENTANGLING CORRELATED VARIABLES

In this section, we introduce a toy disentanglement problem where all quantities of interest can be computed analytically. First, we show that the supervised loss alone does not yield robust disentangled representations, and we discuss why this is problematic. Then, we show that additionally minimizing the unconditional MI forces the model to learn an *even worse solution*. Finally, we show that minimizing the conditional MI yields appropriately disentangled representations that are robust to correlation shift. This analysis motivates CMI as a good objective for achieving robust disentanglement.

**Problem Statement.**    Suppose we observe noisy data $\mathbf{x} \in \mathbb{R}^m$ obtained from an (unknown) generative process $\mathbf{x} = g(\mathbf{s})$ where $\mathbf{s} = (s_1, s_2, \ldots, s_K)$ are the *underlying factors of variation*, also called source variables or attributes, which may be correlated with each other. We wish to find a transformation $f : \mathbb{R}^m \to \mathbb{R}^n$ to a latent space $f(\mathbf{x}) = \mathbf{z} = (\mathbf{z}_1, \mathbf{z}_2, \ldots, \mathbf{z}_K)$ such that each of the original attributes $s_k$ can be recovered from the corresponding subspace $\mathbf{z}_k$ by a linear mapping $\mathbf{R}_k$, e.g., $\hat{s}_k = \mathbf{R}_k \mathbf{z}_k$. We denote by $\mathbf{z}_{-i}$ the set of subspaces $\{\mathbf{z}_1, \ldots, \mathbf{z}_{i-1}, \mathbf{z}_{i+1}, \ldots, \mathbf{z}_K\}$. We consider three different objectives for learning the latent subspaces: 1) minimizing a supervised loss $L$ (e.g., mean squared error or cross-entropy), $\sum_{i=1}^{K} L(\hat{s}_i, s_i)$, denoted *"Base"*; 2) minimizing the *unconditional mutual information between subspaces* in addition to the supervised loss, $\sum_i L(\hat{s}_i, s_i) + I(\mathbf{z}_1, \ldots, \mathbf{z}_K)$, denoted *"Base+MI"*; and 3) minimizing the *conditional mutual information between subspaces conditioned on observed attributes*, in addition to the supervised loss, $\sum_i L(\hat{s}_i, s_i) + I(\mathbf{z}_i; \mathbf{z}_{-i} \mid s_i)$ denoted *"Base+CMI"*.

### 3.1    FULL SUPERVISION DOES NOT YIELD DISENTANGLEMENT

Consider a linear generative model with correlated Gaussian source variables $\mathbf{s}$, given by:

$$\mathbf{x} = \mathbf{As} + \mathbf{n} \quad , \quad \mathbf{s} \sim \mathcal{N}(\mathbf{0}, \mathbf{C_s}) \quad , \quad \mathbf{n} \sim \mathcal{N}(\mathbf{0}, \mathbf{C_n})$$

where $\mathbf{A}$ is the mixing matrix and $\mathbf{C_s}$ and $\mathbf{C_n}$ are the covariance matrices for the source and noise variables, respectively. We assume that $\mathbf{x}$ is observed and wish to disentangle the underlying source variables $\mathbf{s}$; this corresponds to finding the mapping $\mathbf{A}^{-1}$ that inverts the data generating process. When we have access to the source variables, a natural approach is to minimize a supervised loss

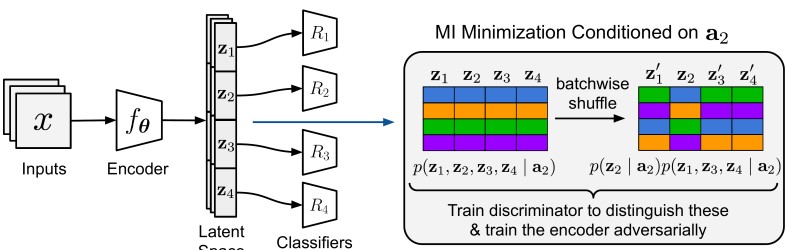

Figure 1: **Adversarial minimization of conditional mutual information via latent-space shuffling.** We minimize the CMI between latent subspaces, $I(\mathbf{z}_1; \cdots; \mathbf{z}_K \mid \mathbf{a}_k)$. Here, we illustrate the algorithm for four attributes with corresponding latent spaces $\{\mathbf{z}_1, \mathbf{z}_2, \mathbf{z}_3, \mathbf{z}_4\}$, where we condition on attribute $\mathbf{a}_2$.

to ensure that each subspace contains information about its attribute. The optimal linear regression solution, both in the least squares sense and with respect to maximum likelihood, is given by the posterior mean: $\hat{\mathbf{s}}(\mathbf{x}) = \mathbb{E}\left[\mathbf{s} \mid \mathbf{x}\right] = \mathbf{C_{sx}C_x^{-1}x}$ where $\mathbf{C_{sx}} = \mathbb{E}\left[\mathbf{s}(\mathbf{As} + \mathbf{n})^{\top}\right] = \mathbf{C_s A^{\top}}$ and $\mathbf{C_x} = \mathbf{A C_s A^{\top}} + \mathbf{C_n}$. The least-squares optimal mapping $\mathbf{C_{sx}C_x^{-1}x}$ is not equal to the inverse $\mathbf{A}^{-1}$ of the generative model, as it is biased by the correlation structure $\mathbf{C_s}$ and $\mathbf{C_n}$ towards directions of maximal signal-to-noise ratio. Thus, regression is sensitive to noise and does not disentangle the underlying sources. In Table 1, we see that in the two-dimensional case where $\mathbf{s} = (s_1, s_2)$ for $\mathbf{A} = \mathbf{I}$, $\sigma = 0.1$ and $\mathrm{corr}(s_1, s_2) = 0.8$, $\hat{\mathbf{s}}$ explains $91.9\%$ of the variance in $\mathbf{s}$ (column *"Base"*). However, when $s_1$ and $s_2$ are uncorrelated, performance drops to $87.6\%$. This drop occurs because the estimator $\hat{\mathbf{s}}$ tries to make use of the assumed correlation between $s_1$ and $s_2$ to counteract the information lost due to noise, but this correlation is no longer present in the test data.

### 3.2 Unconditional Disentanglement Fails Under Correlation Shift

In the 2D linear case, we have:

$$\mathbf{z} = (z_1, z_2) = \mathbf{Wx}, \qquad \widehat{s}_1 = R_1 z_1, \qquad \widehat{s}_2 = R_2 z_2 \tag{1}$$

where the matrix $\mathbf{W}$ encodes the observation into the latent space. In standard supervised objectives, there is no constraint preventing a subspace $z_k$ from containing information about other source variables than $s_k$. A common approach to counter the lack of generalization is to minimize the MI between the latent subspaces $z_1$ and $z_2$ (Chen et al., 2018; Peng et al., 2019). In the Gaussian case, random variables are independent if and only if they are *uncorrelated*. The optimal linear regression weights that yield $I(z_1; z_2) = 0$ (e.g., $\mathbf{W}$ such that $\mathrm{Cov}(\mathbf{z})$ is diagonal) can be computed by whitening $\mathbf{x}$ and rotating the result by an angle $\phi_{\mathrm{opt}}$ which leads to maximal VE ($\phi_{\mathrm{opt}} = -\pi/4$ for positive correlations and $\mathbf{A} = \mathbf{I}$). See Figure 6 in Appendix C for details. However, the resulting model no longer performs well on in-distribution-data (Table 1, column *"Base+MI"*). There is correlation between the sources $s_1$ and $s_2$ and therefore $I(s_1; s_2) > 0$. By enforcing independence, at least one of the subspaces cannot contain all relevant information about its target value and therefore will have poor predictive performance (see Proposition 1 in Appendix E).

### 3.3 Conditional Disentanglement is Robust to Correlation Shift

We have seen that enforcing unconditional independence between the latent spaces does not solve the disentanglement problem. However, from the graphical model (Appendix C, Fig. 5), it is clear that $\mathbf{z}_1$ and $\mathbf{z}_2$ are independent *conditioned on either of $s_1$ or $s_2$*: assuming a common cause for the correlation between $s_1$ and $s_2$, there is a connection in the graphical model between $\mathbf{z}_1$ and $\mathbf{z}_2$ introducing the statistical dependence. Observing either $s_1$ or $s_2$ disconnects $\mathbf{z}_1$ and $\mathbf{z}_2$. Enforcing independence *conditioned on each of the source variables* is also sufficient to yield a robust disentangled representation. For our 2D example, this corresponds to $\mathrm{I}(\mathbf{z}_1; \mathbf{z}_2 \mid s_1) = 0$ and $\mathrm{I}(\mathbf{z}_1; \mathbf{z}_2 \mid s_2) = 0$. If $s_1$ and $s_2$ are correlated and if we can predict $s_1$ from $\mathbf{z}_1$, then it must be the case that $\mathbf{z}_1$ contains information about $s_2$. To improve robustness to shifts, we wish to ensure that $\mathbf{z}_1$ and $\mathbf{z}_2$ share as little information as possible, namely exactly the information which is necessary to account for the correlation between the sources. This can be enforced via $I(\mathbf{z}_1, \mathbf{z}_2 | s_2)$, which states that if we know $s_2$, then knowing $\mathbf{z}_1$ does not give us more information about $\mathbf{z}_2$. The same logic applies for the need of $I(\mathbf{z}_1, \mathbf{z}_2 | s_1)$. The optimal solution under the constraint

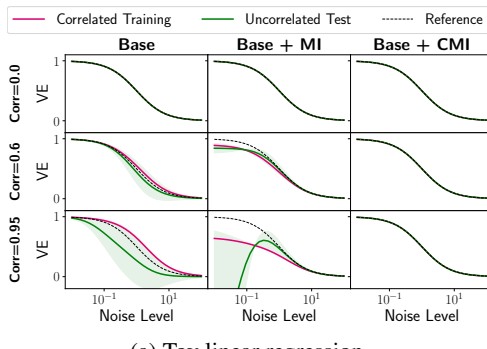
(a) Toy linear regression.

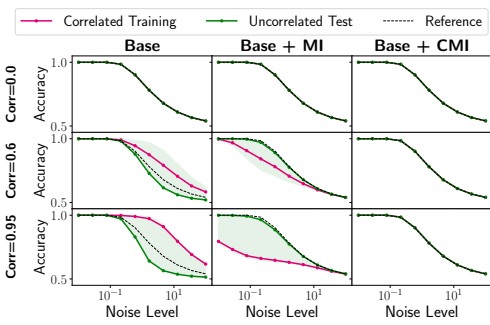
(b) Toy classification with ten attributes.

Figure 2: **Synthetic linear regression and linear classification tasks.** We investigate the impact of the correlation strength and noise level when using each of the objectives *Base*, *Base+MI*, and *Base+CMI*. We measure the variance explained for regression and accuracy for classification. We compare the objectives (columns) with different training correlations (rows). In both tasks, we sweep over noise levels in the range $[10^{-2}, 10^2]$. The base models in the uncorrelated setting serve as a reference (dashed black line). Magenta: performance on correlated training data. Green: performance on test data with a range of correlation shifts (solid line: uncorrelated data; shaded region: correlations in the range $[-1, 1]$ for regression, and only positive correlations for classification). In both regression and classification tasks, we find that *Base+CMI* leads to robustness under correlation shift, while the other approaches do not.

of conditional independence is achieved by the mapping $\mathbf{W} = \mathbf{A}^{-1}$, successfully recovering the underlying generative model. We find that for this solution, the performance of the model is not affected by the correlation shift (Table 1, column *"Base+CMI"*). This demonstrates the usefulness of minimizing CMI and motivates us to investigate CMI minimization for larger-scale tasks.

## 4 MINIMIZING CMI

In simple cases such as linear regression, we can compute and minimize the MI/CMI analytically; however, for most tasks (including classification), there is no closed form for the mutual information. In this section, we describe our approach to minimize the CMI in general classification settings. We formally describe the algorithms for the baselines (*Base* and *Base + MI*) in Appendix B.

Suppose we have a dataset $\mathcal{D} = \{(\mathbf{x}^{(i)}, \mathbf{a}^{(i)})\}_{i=1}^N$ where $\mathbf{x}^{(i)}$ is an example and $\mathbf{a}^{(i)}$ is a vector of attribute labels ($\mathbf{a}_k^{(i)}$ is the label for the $k^{\text{th}}$ attribute of the $i^{\text{th}}$ example). We consider discrete attribute values, $\mathbf{a}_k^{(i)} \in \mathbb{N}$. Let $f_{\boldsymbol{\theta}} : \mathbf{x} \mapsto \mathbf{z}$ denote an encoder function parameterized by $\boldsymbol{\theta}$ that maps examples $\mathbf{x} \in \mathbb{R}^m$ to latent representations $\mathbf{z} \in \mathbb{R}^n$. We aim to learn one latent subspace per attribute, such that each subspace is independent from all other subspaces conditioned on the encoded attribute.

To obtain samples from $p(\mathbf{z}_1, \ldots, \mathbf{z}_K \mid \mathbf{a}_k)$ and $p(\mathbf{z}_k \mid \mathbf{a}_k)p(\mathbf{z}_{-k} \mid \mathbf{a}_k)$, we loop over values of $\mathbf{a}_k$, and for each condition $\{\mathbf{a}_k = 0, \mathbf{a}_k = 1, \ldots\}$, we select examples from the minibatch that satisfy the condition, giving us samples from $p(\mathbf{z}_1, \ldots, \mathbf{z}_K \mid \mathbf{a}_k)$; then we shuffle the latent subspaces $\mathbf{z}_j, \forall j \neq k$ jointly batchwise (e.g., combining $\mathbf{z}_k$ from one example with $\mathbf{z}_{-k}$ from another) to obtain samples from $p(\mathbf{z}_k \mid \mathbf{a}_k)p(\mathbf{z}_{-k} \mid \mathbf{a}_k)$. To enforce $p(\mathbf{z}_1, \ldots, \mathbf{z}_K \mid \mathbf{a}_k) = p(\mathbf{z}_k \mid \mathbf{a}_k)p(\mathbf{z}_{-k} \mid \mathbf{a}_k)$, we train the encoder $f$ adversarially against a discriminator trained to distinguish between these two distributions. The discriminator takes as input a representation and predicts whether it is "real" (e.g., drawn from the joint distribution) or "fake" (e.g., drawn from the product of marginals). One discriminator is trained for each attribute $\mathbf{a}_k$, which receives samples from the two distributions and the attribute value it is conditioned on. In practice, we use a conditional discriminator, effectively sharing parameters between the discriminators for each of the attributes. This approach is architecture-agnostic. The process is illustrated in Figure 1 and the corresponding algorithms are in Appendix B.

## 5 EXPERIMENTS

First, we present results for synthetic regression and classification tasks. Next, investigate a correlated version of CelebA under fully and weakly supervised settings, and show that minimizing CMI can largely eliminate the gap in performance under correlation shift. Experimental details and additional experiments are in Appendix C and D.

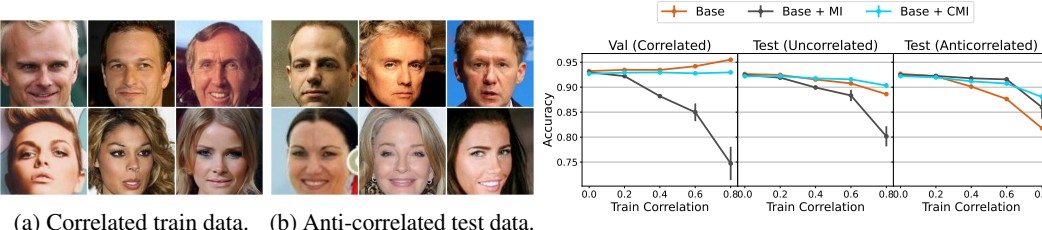

(a) Correlated train data.    (b) Anti-correlated test data.

(c) Performance comparison.

Figure 3: **Correlated CelebA.** (a) Training examples with correlation 0.8 between attributes `Male` and `Smiling`, such that the majority of men are smiling while the majority of women are not. (b) Anti-correlated test examples, where the majority of women are smiling. (c) Accuracies of each method under a range of correlation strengths, for validation data with the same correlation as the training data, uncorrelated test data, and anticorrelated test data.

**Toy Linear Regression.**    In the linear regression problem in Section 3, we evaluated the performance under correlation shift for one specific noise level and correlation strength. Here, we varied these parameters and found that our findings hold over all noise levels and non-zero correlation strengths. As shown in Figure 2a, the performance of *Base* drops most severely under correlation shift for strong correlations and intermediate noise levels. In this regime, the advantage of *Base+CMI* is most clear.

**Toy Multi-Attribute Classification.**    We show in a controlled setting that our findings hold for classification tasks with multiple attributes. Here, the binary source attributes $a_k = \pm 1$, $\forall k \in \{1, \dots, K\}$ generate the observed data via $\mathbf{x} = \mathbf{A}\mathbf{a} + \mathbf{n}$ (we set $\mathbf{A} = \mathbf{I}$ for simplicity) with normally distributed noise $\mathbf{n} \sim \mathcal{N}(\mathbf{0}, \mathbf{C_n})$. We induce pairwise correlations between the attributes $a_k$. As for the regression task, we find that *Base+CMI* leads to robustness under correlation shift (Figure 2b).

**CelebA.**    Finally, we consider a realistic setting using the CelebA faces dataset (Liu et al., 2015). We considered two attributes that we knew *a priori* were not causally related, `Male` and `Smiling`, and we created subsampled datasets that satisfied specific correlations between attributes. We investigated a range of training correlations $\{0, 0.2, 0.4, 0.6, 0.8\}$, and evaluated our models on both *anti-correlated* and *uncorrelated* test sets (Figures 3a and 3b). We found that minimizing CMI has a larger effect for medium-to-high correlation; however, CMI minimization does not hurt performance at low correlation strengths (Figure 3c). Note that while the unconditional model appears to have good performance on the anti-correlated test set, its performance is poor on the validation set (that has the same correlation structure as the training set), so this model does not perform well on in-distribution data. In contrast, the conditional MI model performs well on both in-distribution data and shifted test distributions. Also note that the problem of disentangling correlated attributes does not occur only under correlation shift,

but is already present in the source domain where certain attribute combinations will reliably be treated incorrectly. For example, *Base* fails to recognize the rare non-smiling male faces in 49% of the cases, while *Base+CMI* fails only in 25% of the cases.

**Extension to Weakly Supervised Settings.** Our method can be applied directly to weakly supervised settings. Importantly, it is not necessary to have labels for multiple attributes for a single data point. We find that when reducing the number of labels, *Base+CMI* outperforms the other objectives under correlation shift (Figure 4).

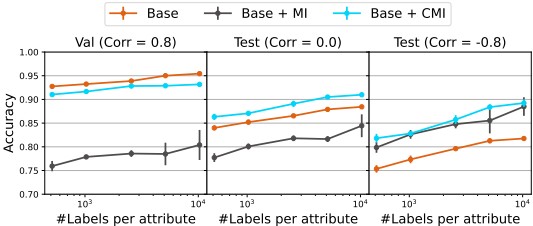

Figure 4: **Weakly-supervised CelebA.** The x-axis shows the number of labels per attribute used during training; the rightmost datapoint corresponds to full supervision. Here, *Base+CMI* outperforms the other objectives under correlation shift.

## 6    CONCLUSION

Correlations are prevalent in real-world data, yet pose a substantial challenge for disentangled representation learning. Standard approaches learn to rely on these correlations, and when the attributes are not causally related, this leads to poor performance under test-time correlation shift. We establish CMI minimization as a more appropriate alternative to mutual information minimization, which sets the stage for the development of more powerful objective functions for disentanglement.

ACKNOWLEDGEMENTS

We thank Jörn-Henrik Jacobsen for his valuable contributions in the early stage of this work. We thank Steffen Schneider, Dylan Paiton, Lukas Schott, Elliot Creager, and Frederik Träuble for helpful discussions. We thank the International Max Planck Research School for Intelligent Systems (IMPRS-IS) for supporting Christina Funke. Paul Vicol was supported by a JP Morgan AI Fellowship.

We acknowledge support from the German Federal Ministry of Education and Research (BMBF) through the Competence Center for Machine Learning (FKZ 01IS18039A) and the Bernstein Computational Neuroscience Program Tübingen (FKZ: 01GQ1002), the German Excellence Initiative through the Centre for Integrative Neuroscience Tübingen (EXC307), and the Deutsche Forschungsgemeinschaft (DFG; Projektnummer 276693517 – SFB 1233). Resources used in preparing this research were provided, in part, by the Province of Ontario, the Government of Canada through CIFAR, and companies sponsoring the Vector Institute www.vectorinstitute.ai/partners.

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

## A    EXTENDED RELATED WORK

**Estimating & Optimizing Mutual Information.**    Many approaches have been proposed for MI and CMI estimation and optimization. The Mutual Information Neural Estimator (MINE) (Belghazi et al., 2018) uses a lower-bound of the MI based on the Donsker-Varadhan dual representation of the KL divergence (Donsker & Varadhan, 1983). Poole et al. (2019) provide an overview of variational bounds that can be used to estimate MI; most are *lower bounds*, which are useful in principle for *maximizing* MI, but which have also been used to minimize MI (even though minimizing a lower bound is not guaranteed to decrease MI). CLUB (Cheng et al., 2020) introduced a variational upper bound of MI, providing a more principled objective for minimizing MI. Several CMI estimators have been proposed, including conditional-MINE (Molavipour et al., 2020a), C-MI-GAN (Mondal et al., 2020), CCMI (Mukherjee et al., 2020), and an approach based on nearest neighbors (Molavipour et al., 2020b). Many approaches to MI minimization are based on batchwise shuffling of latent subspaces, sometimes referred to as metameric sampling (Belghazi et al., 2018; Nemeth, 2020; Feng et al., 2018; Park et al., 2020; Peng et al., 2019). The approach we use in Section 4 follows this paradigm of latent-space shuffling.

**Correlations Between Features.**    With roots in ICA, most research on disentanglement focuses on data that was generated by independent factors, including synthetic benchmarks such as dSprites (Matthey et al., 2017), Shapes3D (Burgess & Kim, 2018), Cars3D (Reed et al., 2015), SmallNORB (LeCun et al., 2004), or MPI3D (Gondal et al., 2019). In real-world datasets on the other hand, factors are often correlated (Welinder et al., 2010; Lin et al., 2014). Träuble et al. (2020) pointed out the challenges that arise when attempting to learn disentangled representations on correlated data, and performed a large-scale empirical evaluation of the effect of correlations on widely-used VAE-based disentanglement models. Our work shows that even under full supervision, correlated attributes are problematic when enforcing independence between latent subspaces. Causally-informed modeling (Zhang et al., 2020) is another approach to learning disentangled representations and extracting invariant features; an example of this is Invariant Risk Minimization (IRM) (Arjovsky et al., 2019). To investigate the effect of correlations systematically, it is common to modify existing datasets to induce correlations, for example by subsampling the data, or by generating synthetic datasets with the desired properties (Dittadi et al., 2020; Cimpoi et al., 2014; Jacobsen et al., 2018; Locatello et al., 2019b). We follow this approach in our experiments.

**Fairness.**    An important application of disentanglement is fairness. Typically, this can be addressed by modifying the training data to be unbiased or by adding a regularizer (e.g. based on mutual information) that quantifies and minimizes the degree of bias (Kamiran & Calders, 2009; Kamishima et al., 2011; Zemel et al., 2013; Hardt et al., 2016; Cho et al., 2020).

**Mutual Information.**    The mutual information (MI) between two random variables $\mathbf{x}$ and $\mathbf{y}$, denoted $I(\mathbf{x}; \mathbf{y})$, is the KL divergence between the joint distribution $p(\mathbf{x}, \mathbf{y})$ and the product of the marginal distributions $p(\mathbf{x})p(\mathbf{y})$: $I(\mathbf{x}; \mathbf{y}) = D_{\mathrm{KL}}[p(\mathbf{x}, \mathbf{y})||p(\mathbf{x})p(\mathbf{y})]$. MI minimization is at the heart of many approaches to disentanglement. The *conditional mutual information* (CMI) is defined as:

$$I(\mathbf{x}; \mathbf{y} \mid \mathbf{z}) = \mathbb{E}_{\mathbf{z}} \left[ D_{\mathrm{KL}}[p(\mathbf{x}, \mathbf{y} \mid \mathbf{z}) \,||\, p(\mathbf{x} \mid \mathbf{z})p(\mathbf{y} \mid \mathbf{z})] \right]$$

CMI measures the dependency between two variables given that we know the value of a third variable. For example, there is a dependency between a country's number of Nobel laureates per capita and chocolate consumption per capita (Prinz, 2020). However, this dependency is largely explained by the wealth of a country, thus $I(nobel; chocolate \mid wealth) < I(nobel; chocolate)$. In general, the CMI can be smaller or larger than the unconditional MI.

## B    ALGORITHMS

In this section, we provide formal descriptions of the algorithms we consider in this paper. Algorithm 1 describes the classification-only baseline, that trains separate linear classifiers to predict attributes $\mathbf{a}_k$ from the corresponding latent subspaces $\mathbf{z}_k$ (corresponds to *"Base"*). Algorithm 2 and Algorithm 3 describe the encoder and discriminator training loops for the unconditional disentanglement baseline (corresponds to *"Base+MI"*). This objective adversarially minimizes the discrepancy between samples from the joint distribution $p(\mathbf{z}_1, \ldots, \mathbf{z}_k)$ and the product of marginals

$p(\mathbf{z}_1) \cdots p(\mathbf{z}_k)$. Algorithm 4 and Algorithm 5 describe the encoder and discriminator training loops for conditional disentanglement (corresponds to *"Base+CMI"*).

---

**Algorithm 1** Supervised Learning on Subspaces

---

1: **Input:** $\{\phi_1, \ldots, \phi_K\}$, initial parameters for $K$ linear classifiers $C_1, \ldots, C_K$
2: **Input:** $\theta$, initial parameters for the encoder $F$
3: **Input:** $\alpha, \beta$ learning rates for training the encoder and linear classifiers
4: **while** true **do**
5: $\quad (\mathbf{x}, \{\mathbf{a}_k\}_{k=1}^K) \sim \mathcal{D}_{\text{Train}}$ $\qquad\qquad\qquad$ ▷ Sample a minibatch of data with attribute labels
6: $\quad \mathbf{z} \leftarrow F_\theta(\mathbf{x})$ $\qquad\qquad\qquad\qquad\qquad\qquad$ ▷ Forward pass through the encoder
7: $\quad \{\mathbf{z}_k\}_{k=1}^K \leftarrow \text{SplitSubspaces}(\mathbf{z}, k)$ $\qquad$ ▷ Partition the latent space into $k$ subspaces
8: $\quad L \leftarrow \sum_{k=1}^K L_{\text{cls}}(C_k(\mathbf{z}_k; \phi_k), \mathbf{a}_k)$ $\qquad\qquad$ ▷ Cross-entropy for each attribute
9: $\quad \theta \leftarrow \theta - \alpha \nabla_\theta L$ $\qquad\qquad\qquad\qquad\qquad$ ▷ Update encoder parameters
10: $\quad \phi_k \leftarrow \phi_k - \beta \nabla_{\phi_k} L \quad , \quad \forall k \in \{1, \ldots, K\}$ $\qquad$ ▷ Update classifier parameters
11: **end while**

---

**Algorithm 2** Learning Unconditionally Disentangled Subspaces — Training the Encoder

---

1: **Input:** $\{\phi_1, \ldots, \phi_K\}$, initial parameters for $K$ linear classifiers $C_1, \ldots, C_K$
2: **Input:** $\theta$, initial parameters for the encoder $F$
3: **Input:** $\alpha, \beta$ learning rates for training the encoder and linear classifiers
4: **while** true **do**
5: $\quad (\mathbf{x}, \{\mathbf{a}_k\}_{k=1}^K) \sim \mathcal{D}_{\text{Train}}$ $\qquad\qquad\qquad$ ▷ Sample a minibatch of data with attribute labels
6: $\quad \mathbf{z} \leftarrow F_\theta(\mathbf{x})$ $\qquad\qquad\qquad\qquad\qquad\qquad$ ▷ Forward pass through the encoder
7: $\quad \{\mathbf{z}_k\}_{k=1}^K \leftarrow \text{SplitSubspaces}(\mathbf{z}, k)$ $\qquad$ ▷ Partition the latent space into $k$ subspaces
8: $\quad L \leftarrow \sum_{k=1}^K L_{\text{cls}}(C_k(\mathbf{z}_k; \phi_k), \mathbf{a}_k)$ $\qquad\qquad$ ▷ Cross-entropy for each attribute
9: $\quad \mathbf{z}' \sim p(\mathbf{z}_1) p(\mathbf{z}_2) \cdots p(\mathbf{z}_k)$ $\qquad\qquad$ ▷ Samples w/ batchwise-shuffled subspaces
10: $\quad L \leftarrow L + \log(1 - D_\omega(\mathbf{z}')) + \log(D_\omega(\mathbf{z}))$ $\qquad\qquad$ ▷ Add adversarial loss
11: $\quad \theta \leftarrow \theta - \alpha \nabla_\theta L$ $\qquad\qquad\qquad\qquad\qquad$ ▷ Update encoder parameters
12: $\quad \phi_k \leftarrow \phi_k - \beta \nabla_{\phi_k} L \quad , \quad \forall k \in \{1, \ldots, K\}$ $\qquad$ ▷ Update classifier parameters
13: **end while**

---

**Algorithm 3** Learning Unconditionally Disentangled Subspaces — Training the Discriminator

---

1: **Input:** $\omega$, initial parameters for the discriminator $D$
2: **Input:** $\gamma$, learning rate for training the discriminator
3: **while** true **do**
4: $\quad (\mathbf{x}, \{\mathbf{a}_k\}_{k=1}^K) \sim \mathcal{D}_{\text{Train}}$ $\qquad\qquad\qquad$ ▷ Sample a minibatch of data with attribute labels
5: $\quad \mathbf{z} \leftarrow F_\theta(\mathbf{x})$ $\qquad\qquad\qquad\qquad\qquad\qquad$ ▷ Forward pass through the encoder
6: $\quad \{\mathbf{z}_k\}_{k=1}^K \leftarrow \text{SplitSubspaces}(\mathbf{z}, k)$ $\qquad$ ▷ Partition the latent space into $k$ subspaces
7: $\quad \mathbf{z}' \sim p(\mathbf{z}_1) p(\mathbf{z}_2) \cdots p(\mathbf{z}_k)$ $\qquad\qquad$ ▷ Samples w/ batchwise-shuffled subspaces
8: $\quad L \leftarrow L + \log(D_\omega(\mathbf{z}')) + \log(1 - D_\omega(\mathbf{z}))$ $\qquad\qquad$ ▷ Add adversarial loss
9: $\quad \omega \leftarrow \omega - \gamma \nabla_\omega L$ $\qquad\qquad\qquad\qquad\qquad$ ▷ Update discriminator parameters
10: **end while**

---

---

**Algorithm 4** Learning Conditionally Disentangled Subspaces Adversarially — Training the Encoder

---

1: **Input:** $\{\phi_1, \ldots, \phi_K\}$, initial parameters for $K$ linear classifiers $R_1, \ldots, R_K$
2: **Input:** $\boldsymbol{\theta}$, initial parameters for the encoder $f$
3: **Input:** $\alpha, \beta$ learning rates for training the encoder and linear classifiers
4: **while** true **do**
5: $\quad (\mathbf{x}, \{\mathbf{a}_k\}_{k=1}^K) \sim \mathcal{D}_{\text{Train}}$ $\qquad\qquad$ ▷ Sample a minibatch of data with attribute labels
6: $\quad \mathbf{z} \leftarrow f_{\boldsymbol{\theta}}(\mathbf{x})$ $\qquad\qquad\qquad\qquad$ ▷ Forward pass through the encoder
7: $\quad \{\mathbf{z}_k\}_{k=1}^K \leftarrow \text{SplitSubspaces}(\mathbf{z}, K)$ $\qquad$ ▷ Partition the latent space into $K$ subspaces
8: $\quad L \leftarrow \sum_{k=1}^K L_{\text{cls}}(R_k(\mathbf{z}_k; \phi_k), \mathbf{a}_k)$ $\qquad$ ▷ Cross-entropy for each attribute
9: $\quad$ **for** $k \in \{1, \ldots, K\}$ **do** $\qquad\qquad\qquad$ ▷ For each attribute/subspace
10: $\qquad \mathbf{z}' \sim p(\mathbf{z}_1, \ldots \mathbf{z}_K \mid \mathbf{a}_k)$ $\qquad\qquad$ ▷ Samples from the joint distribution
11: $\qquad \mathbf{z}'' \sim p(\mathbf{z}_k \mid \mathbf{a}_k) p(\mathbf{z}_{-k} \mid \mathbf{a}_k)$ $\qquad$ ▷ Samples w/ batchwise-shuffled subspaces
12: $\qquad L \leftarrow L + \log\left(1 - D_{\boldsymbol{\omega}}(\mathbf{z}'')\right) + \log\left(D_{\boldsymbol{\omega}}(\mathbf{z}')\right)$ $\qquad$ ▷ Add adversarial loss
13: $\quad$ **end for**
14: $\quad \boldsymbol{\theta} \leftarrow \boldsymbol{\theta} - \alpha \nabla_{\boldsymbol{\theta}} L$ $\qquad\qquad\qquad$ ▷ Update encoder parameters
15: $\quad \phi_k \leftarrow \phi_k - \beta \nabla_{\phi_k} L \quad , \quad \forall k \in \{1, \ldots, K\}$ $\qquad$ ▷ Update classifier parameters
16: **end while**

---

**Algorithm 5** Learning Conditionally Disentangled Subspaces Adversarially – Training the Discriminator

---

1: **Input:** $\boldsymbol{\omega}$, initial parameters for the discriminator $D$
2: **Input:** $\gamma$, learning rate for training the discriminator
3: **while** true **do**
4: $\quad (\mathbf{x}, \{\mathbf{a}_k\}_{k=1}^K) \sim \mathcal{D}_{\text{Train}}$ $\qquad\qquad$ ▷ Sample a minibatch of data with attribute labels
5: $\quad \mathbf{z} \leftarrow F_{\boldsymbol{\theta}}(\mathbf{x})$ $\qquad\qquad\qquad\qquad$ ▷ Forward pass through the encoder
6: $\quad \{\mathbf{z}_k\}_{k=1}^K \leftarrow \text{SplitSubspaces}(\mathbf{z}, k)$ $\qquad$ ▷ Partition the latent space into $K$ subspaces
7: $\quad L \leftarrow 0$ $\qquad\qquad\qquad\qquad$ ▷ $L$ will accumulate the losses over all subspaces
8: $\quad$ **for** $k \in \{1, \ldots, K\}$ **do**
9: $\qquad \mathbf{z}' \sim p(\mathbf{z}_1, \ldots \mathbf{z}_K \mid \mathbf{a}_k)$ $\qquad\qquad$ ▷ Samples from the joint distribution
10: $\qquad \mathbf{z}'' \sim p(\mathbf{z}_k \mid \mathbf{a}_k) p(\mathbf{z}_{-k} \mid \mathbf{a}_k)$ $\qquad$ ▷ Samples w/ batchwise-shuffled subspaces
11: $\qquad L \leftarrow L + \log\left(D_{\boldsymbol{\omega}}(\mathbf{z}'')\right) + \log\left(1 - D_{\boldsymbol{\omega}}(\mathbf{z}')\right)$ $\qquad$ ▷ Add adversarial loss
12: $\quad$ **end for**
13: $\quad \boldsymbol{\omega} \leftarrow \boldsymbol{\omega} - \gamma \nabla_{\boldsymbol{\omega}} L$ $\qquad\qquad\qquad$ ▷ Update discriminator parameters
14: **end while**

---

## C EXPERIMENTAL DETAILS

**Graphical Model.** In Figure 5 we provide the graphical model discussed in Section 3.3.

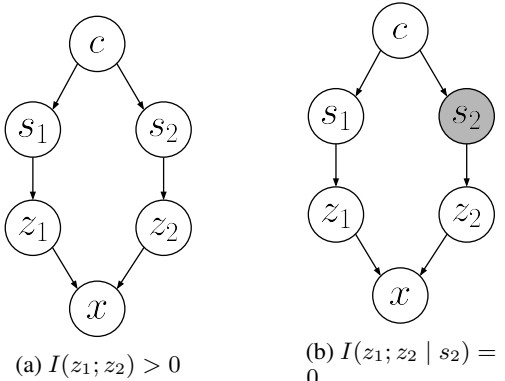

(a) $I(z_1; z_2) > 0$

(b) $I(z_1; z_2 \mid s_2) = 0$

Figure 5: The graphical model for two sources $s_1, s_2$ and corresponding latent subspaces $z_1, z_2$. We assume the source variables have a common cause $c$. In (a), when none of the sources are observed, there is a path from $z_1$ to $z_2$, so we have $I(z_1; z_2) > 0$; in (b) we observe $s_2$, which breaks the path, and thus $I(z_1; z_2 \mid s_2) = 0$.

**Compute Environment.** Our experiments were implemented using PyTorch (Paszke et al., 2019), and were run on our internal clusters. The toy 2D experiments were run on a single NVIDIA RTX 2080 TI GPU, and took approximately 48 hours for all the results presented. The MNIST and CelebA experiments were run on NVIDIA Titan Xp GPUs. Each run of the multi-digit MNIST and CelebA tasks for a given method and correlation strength (and noise level in the MNIST case) took approximately 12 hours, and these were run in parallel.

**Toy Linear Regression.** Here, we describe the optimal solution for the *Base+MI* objective for the linear regression task with $\mathbf{A} = \mathbf{I}$. Optimal linear regression with zero mutual information between $z_1$ and $z_2$ can be obtained by taking the singular value decomposition of $\mathbf{x}$ followed by whitening and rotation by $-45°$ (see Figure 6).

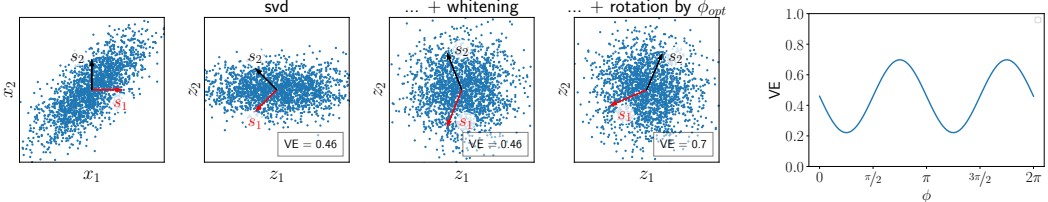

Figure 6: To enforce unconditional independence, we choose $\mathbf{W}$ such that $\mathrm{Cov}(\mathbf{z})$ is diagonal. In our case this is easy: the principal components of $\mathbf{x}$ are $x_1 + x_2$ and $x_1 - x_2$. The optimal regression loss with minimal MI is then given by whitening and rotating the result by angle $\phi_{\mathrm{opt}}$ which leads to maximal VE ($\phi_{\mathrm{opt}} = -\pi/4$ for positive correlations and $\mathbf{A} = \mathbf{I}$).

**Toy Multi-Attribute Classification.** The following figure illustrates the data $\mathbf{x}$ for different correlation strength and noise levels (Figure 7) in the case of two attributes. In the 2D case increasing the correlation strength means that data points with $a_1 = a_2$ are increasingly more common relative to $a_1 \neq a_2$. In the multi-attribute setting, the correlation strength refers to the pairwise correlation between all attributes.

We used a PacGAN-style setup (Lin et al., 2018) for our toy experiments, where the discriminator takes as input a concatenation of 50 samples.

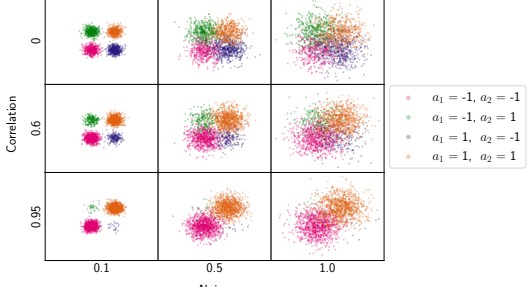

Figure 7: Data used for linear classification with two attributes ($a_1$ and $a_2$), visualized for a range of correlation strengths and noise levels.

- **Base**: We used Adam (Kingma & Ba, 2014) with a learning rate of 0.01.
- **Base + MI**: We used Adam to optimize the encoder, linear classifiers, and discriminators. After each step of optimizing the discriminator and encoder, we optimized the linear classifiers ($R$) for 10 steps. The disentanglement loss term was weighted by a factor of 100 relative to the classification loss. In preliminary tests, we found that the optimal learning rate depended on noise level, correlation strength, and number of attributes. The results in Figure 2b were obtained using one of the following learning rates for the discriminator $\{1e-4, 2e-4, 5e-4, 1e-3, 5e-3\}$. The learning rate of the generator and linear classifiers was chosen to be 10 times smaller than the discriminator learning rate.
- **Base + CMI**: For $\mathbf{A} = \mathbf{I}$, no optimization was necessary, as we already know the optimal solution to be $\mathbf{W} = \mathbf{A}^{-1} = \mathbf{I}$. We confirmed experimentally that the discriminator could not get above chance performance for this solution.

**CelebA.** For all experiments, we used minibatch size 100, and latent dimension $D = 10$. As the encoder model, we used a three-layer MLP with 50 hidden units per layer and ReLU activations. Similarly to the MNIST setup, we trained for 200 epochs, using Adam to optimize the encoder, linear classifiers, and discriminators. For each method, we performed a grid search over learning rates $\{1e-5, 1e-4, 1e-3\}$ separately for each of the encoder, discriminator(s), and linear classification heads; we selected the best learning rates based on validation accuracy.

**Weakly Supervised Setting.** For the fully supervised CelebA experiment, labels for both attributes were available for all 10260 images. For the weakly supervised setting, we reduced the number of labels to 5130 (50% of the labels of the fully supervised dataset), 2565 (25%), 1026 (10%), or 513 (5%) for each attribute. This implies that some images had both labels, some had only one label and some images had no labels (for example when using 50% of the labels the distinction is as follows: 25% of the images had both labels; 25% had only labels for attribute 1; 25% had only labels for attribute 2; and 25% had no labels). The three objectives can be applied to these weakly supervised settings. For *Base*, the cross-entropy loss for each attribute was computed only for the images that had labels for the corresponding attribute. For *Base+MI* no labels are required for the unconditional shuffling; thus this objective can be applied even for the images without labels. For *Base+CMI*, our method shuffles only images that have the same value for a given attribute. This also works if the labels of the other attribute are missing. We used the same training parameters as for the supervised experiment, except for increasing the number of training epochs (up to 1200 epochs) and adapting the minibatch size to the number of labels.

## D  EXTRA EXPERIMENTS

**Extended Analysis for Toy Linear Regression.** Here, we provide intuitive explanations of the behaviors of the different objectives. Comparing the correlation of target $\mathbf{s}$ and data $\mathbf{x}$ with the correlation of the predictions $\hat{\mathbf{s}}$ of the different models can help us understand the findings of Section 3. As shown in Figure 8, the predictions of optimal linear regression (*Base+MI*) ($\mathrm{corr}(\hat{s}_1, \hat{s}_2) = 0.85$) are stronger correlated than the data ($\mathrm{corr}(x_1, x_2) = 0.73$). This shows that the correlation present in the training data is used to compensate for the noise. Enforcing unconditional independence (*Base+MI*) on the other hand, leads to uncorrelated predictions ($\mathrm{corr}(\hat{s}_1, \hat{s}_2) = 0$). Undoubtedly, this cannot be the correct solution, as the targets are correlated. Additionally, when the correlations change, the independence constraint does not hold anymore. This can lead to interesting effects under correlation shift. While for most noise levels, the performance on the test data is poor, for some noise levels the performance can be even higher than training performance (Figure 2a). In these cases the model can "accidentally" exploit the correlation in the test data to make the correct predictions. The correlation of the predictions ($\mathrm{corr}(\hat{s}_1, \hat{s}_2) = 0.73$) matches the correlation of the data only for *Base+CMI*.

**Multi-Digit Occluded MNIST.** As an additional experiment, we designed a larger-scale task to systematically investigate whether the properties found in the toy classification task hold in a more complex setting. We created a dataset by concatenating two MNIST digits side-by-side, where the aim is to predict both the left- and right-hand-side labels. We used a subset of MNIST consisting of classes 3 and 8 (which are visually similar and can become ambiguous under occlusions). We generated

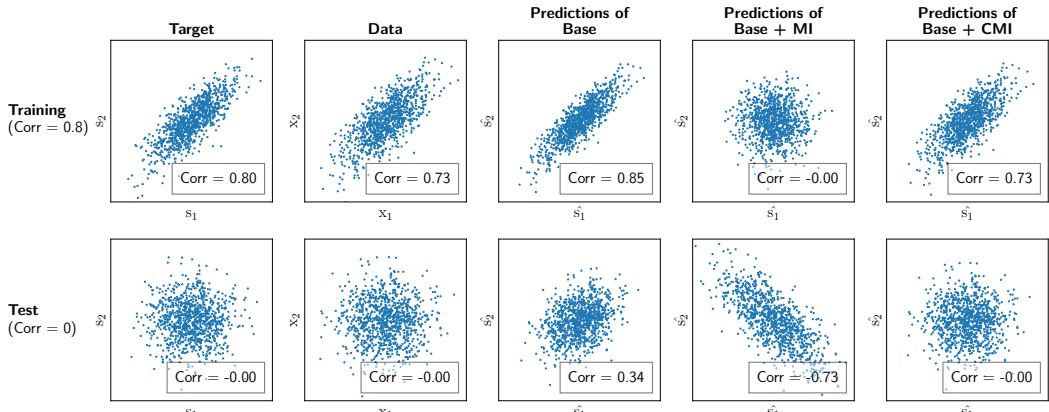

Figure 8: Correlation of target $\mathbf{s}$, data $\mathbf{x}$ and predictions $\hat{\mathbf{s}}$. Only for conditional independence does the correlation between the predictions and data match for both training and test data. The same setup as for Table 1 was used (two attributes $s_1$ and $s_2$, $\text{corr}(s_1, s_2) = 0.8$, $\mathbf{A} = \mathbf{I}$, $\sigma = 0.1$).

occlusion masks using the procedure used by Chai et al. (2021); examples from our synthetic dataset under a range of noise settings are shown in Figure 9a. This mimics the multiple-object classification setting in a way that allows us to control the correlation strength and noise level (via the amount of occlusion), allowing for systematic analysis. While this would also be possible for colored MNIST and dSprites, one advantage of our choice is the symmetry of our task, which allows us to exclude potential side-effects: here, the attributes have the same type (the digit identity), whereas the attributes in colored MNIST (digit identity and color) and dSprites (shape, size, position, etc.) are more diverse.

Similarly to the toy tasks, we train an encoder to map images onto a $D$-dimensional latent space, which is partitioned in two equal-sized subspaces corresponding to the two digits; we train a linear classifier on each subspace to predict the respective class labels. We consider different correlation strengths between the left and right digits in the training set (where strong correlation means that the digits match most of the time, e.g., 3-3 or 8-8 are more common than 3-8 or 8-3). We evaluate each model on test data with correlation strengths ranging from $[-1, 1]$. The results are shown in Figure 9b. We found that the conclusions from the toy experiments hold in this setting: supervised learning with only the classification loss, as well as with unconditional MI minimization, fails under test-time correlation shift, while the model minimizing conditional MI is more robust.

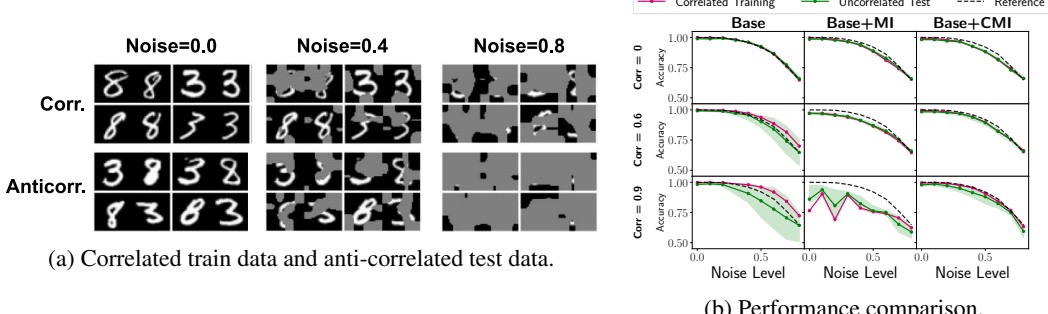

(a) Correlated train data and anti-correlated test data.

(b) Performance comparison.

Figure 9: **Multi-digit occluded MNIST.** (a) Occlusions are generated for each of the left and right digits separately. We visualize both correlated training data (where 3-3 and 8-8 pairs are frequent) and anticorrelated test data (where 3-8 and 8-3 pairs are frequent), under a range of occlusion strengths. (b) Accuracies under correlation shifts for different noise levels, achieved by training with each of the objective functions *Base*, *Base+MI*, and *Base+CMI*.

**Disentanglement Metrics.** Previous work (Locatello et al., 2020a) has shown that common disentanglement metrics are not suitable for the correlated setting. For this reason, we focused on comparing performance under correlation shift, which we consider more suitable for correlated data: if a model cannot predict a factor of variation well for certain values of another factor, then the model did not successfully disentangle these factors of variation. However, one can still make use of the

disentanglement metrics by evaluating them on *uncorrelated data* using the models trained on correlated data. We performed this analysis for two of our datasets and found in both cases that *Base+CMI* reached better scores compared to the other objectives for almost all metrics. Disentanglement results for the toy classification task with ten attributes are shown in Table 2. The disentanglement results for CelebA are shown in Table 3. Since the disentanglement metrics require that the factors of variation are each encoded in one-dimensional subspaces, we set latent dimension $D = 2$ for this experiment.

| Metric | Base | Base+MI | Base+CMI |
|---|---|---|---|
| IRS (Suter et al., 2019) ↑ | 0.377 | 0.573 | **0.605** |
| SAP (Kumar et al., 2017) ↑ | 0.118 | 0.470 | **0.477** |
| MIG (Chen et al., 2018) ↑ | 0.179 | 0.939 | **0.975** |
| DCI Disentanglement (Eastwood & Williams, 2018) ↑ | 0.413 | 0.980 | **0.998** |
| Beta-VAE (Higgins et al., 2017) ↑ | 0.996 | 1 | 1 |
| Factor-VAE (Kim & Mnih, 2018) ↑ | 1 | 1 | 1 |
| Gaussian Total Correlation ↓ | 10.073 | 0.485 | **0.025** |
| Gaussian Wasserstein Corr ↓ | 12.905 | 0.373 | **0.027** |
| Gaussian Wasserstein Corr Norm ↓ | 0.866 | 0.037 | **0.002** |
| Mutual Info Score ↓ | 0.975 | 0.197 | **0.149** |

Table 2: **Disentanglement metrics for toy classification with ten attributes.** Metrics are evaluated on the uncorrelated test set. Bold font indicates model with best disentanglement score.

| Metric | Base | Base+MI | Base+CMI |
|---|---|---|---|
| IRS ↑ | $0.524 \pm 0.043$ | $\mathbf{0.548 \pm 0.038}$ | $0.531 \pm 0.041$ |
| SAP ↑ | $0.306 \pm 0.003$ | $0.296 \pm 0.046$ | $\mathbf{0.389 \pm 0.005}$ |
| MIG ↑ | $0.506 \pm 0.01$ | $0.455 \pm 0.074$ | $\mathbf{0.674 \pm 0.007}$ |
| DCI Disentanglement ↑ | $0.46 \pm 0.009$ | $0.596 \pm 0.038$ | $\mathbf{0.807 \pm 0.023}$ |
| Beta-VAE ↑ | $1.0 \pm 0.0$ | $1.0 \pm 0.0$ | $\mathbf{1.0 \pm 0.0}$ |
| Factor-VAE ↑ | $1.0 \pm 0.0$ | $0.999 \pm 0.003$ | $\mathbf{1.0 \pm 0.0}$ |
| Gaussian Total Correlation ↓ | $0.222 \pm 0.012$ | $0.056 \pm 0.061$ | $\mathbf{0.011 \pm 0.003}$ |
| Gaussian Wasserstein Corr ↓ | $0.351 \pm 0.039$ | $0.01 \pm 0.009$ | $\mathbf{0.002 \pm 0.001}$ |
| Gaussian Wasserstein Corr Norm ↓ | $0.098 \pm 0.005$ | $0.006 \pm 0.004$ | $\mathbf{0.005 \pm 0.001}$ |
| Mutual Info Score ↓ | $0.302 \pm 0.022$ | $0.111 \pm 0.052$ | $\mathbf{0.042 \pm 0.006}$ |

Table 3: **Disentanglement metrics for CelebA.** Metrics are evaluated on the uncorrelated test set. Bold font indicates model with best disentanglement score.

# E PROOF OF PROPOSITION RELATED TO SECTION 3.2

In Section 3.2 we argue that by enforcing independence, at least one of the subspaces cannot contain all relevant information about its target value and therefore will have poor predictive performance. We make this precise in the following proposition:

**Proposition 1** *If $I(s_1; s_2) > 0$, then enforcing $I(z_1; z_2) = 0$ means that $I(z_k; s_k) < H(s_k)$ for at least one $k$.*

*Proof.* Assume that $I(s_1; s_2) > 0$ and at the same time $I(z_k; s_k) = H(s_k)$ (i.e., we are proving by contradiction). Since $I(z_1; s_1) = H(s_1)$, we have $H(s_1 \mid z_1) = 0$ and with $H(s_1 \mid z_1) = H(s_1 \mid z_1, s_2) + I(s_1; s_2 \mid z_1)$ (both non-negative), it follows that $H(s_1 \mid z_1, s_2) = I(s_1; s_2 \mid z_1) = 0$. Since for the interaction information, by definition $I(s_1; s_2; z_1) = I(s_1; s_2) - I(s_1; s_2 \mid z_1)$, and $I(s_1; s_2 \mid z_1) = 0$, we have $I(s_1; s_2; z_1) = I(s_1; s_2) > 0$. Since we also assume $H(s_2 \mid z_2) = 0$, we also have $I(s_1; s_2; z_2) = I(s_1; s_2) > 0$.

We can use this to compute the fourth order interaction information $I(s_1; s_2; z_1; z_2)$. By definition, we have $I(s_1; s_2; z_1; z_2) = I(s_1; s_2; z_1) - I(s_1; s_2; z_1 \mid z_2)$. We just showed that $I(s_1; s_2; z_1) =$

$I(s_1; s_2)$, and therefore we have $I(s_1; s_2; z_1 \mid z_2) = I(s_1; s_2 \mid z_2)$. Together it follows that:

$$I(s_1; s_2; z_1; z_2) = I(s_1; s_2; z_1) - I(s_1; s_2; z_1 \mid z_2) \tag{2}$$
$$= I(s_1; s_2) - I(s_1; s_2 \mid z_2) \tag{3}$$
$$= I(s_1; s_2; z_2) \tag{4}$$
$$= I(s_1; s_2) > 0 \tag{5}$$

On the other hand, we know that $0 = H(s_1 \mid z_1) = H(s_1 \mid z_1; z_2) + I(s_1, z_2 \mid z_1)$ and therefore $I(s_1, z_2 \mid z_1) = 0$. Therefore, the interaction information $I(s_1; z_2; z_1) = I(s_1; z_2) - I(s_1; z_2 \mid z_1) = I(s_1; z_2) \geq 0$. At the same time, we assumed that $I(z_1; z_2) = 0$ and hence $I(z_1; z_2; s_1) + I(z_1; z_2 \mid s_1) = 0$, which shows that $I(z_1; z_2; s_1) \leq 0$. Together, we see that $I(z_1; z_2; s_1) = I(s_1; z_2) = 0$.

Now we can decompose $I(s_1; s_2; z_1; z_2)$ in a different way: $I(s_1; s_2; z_1; z_2) = I(s_1; z_1; z_2) - I(s_1; z_1; z_2 \mid s_2)$. We know that $I(s_1; z_1; z_2) = I(s_1; z_2)$ and therefore $I(s_1; z_1; z_2 \mid s_2) = I(s_1; z_2 \mid s_2) > 0$ and that $I(s_1; z_1; z_2) = 0$. Therefore, it follows that:

$$I(s_1; s_2; z_1; z_2) = I(s_1; z_1; z_2) - I(s_1; z_1; z_2 \mid s_2) \tag{6}$$
$$= 0 - I(s_1; z_2 \mid s_2) \tag{7}$$
$$\leq 0 \tag{8}$$

which is a contradiction with $I(s_1; s_2; z_1; z_2) = I(s_1; s_2) > 0$. Therefore, if $I(s_1; s_2) > 0$ and $I(z_1; z_2) = 0$, it must hold that $I(z_k; s_k) < H(s_k)$ for at least one $k$, which we wanted to show. $\quad\square$

