# OpenReview forum: "Disentanglement and Generalization Under Correlation Shifts"
_ICLR.cc/2022/Workshop/OSC — ICLR2022 OSC  Oral_

### Official Review · Reviewer_ZNS4 · 2022-03-14
**Review: Disentanglement and Generalization Under Correlation Shifts**

**Rating:** 3
**Confidence:** 2

**Review:**

**Summary**: The authors propose minimization of conditional mutual information as a method for learning disentangled representations which generalize more favorably to test distributions with shifts in correlation of the ground truth generative factors. They provide a convincing theoretical argument for the proposed idea based on an analysis of the simple 2-dimensional setting, and further validate this argument with strong empirical results on toy tasks. They further introduce a more general architecture-agnostic training algorithm based on latent-space shuffling which could allow for attribute-conditional mutual information minimization in more realistic settings. They then demonstrate preliminary positive results with this algorithm on a modified version of the Celeb-A dataset where they artificially introduce correlation between attributes.

**Strong points**:
- The approach is well motivated and I believe the preliminary analysis beginning from the low-dimensional linear setting should be highly valued especially the context of this  workshop.
- The extended related work section is thorough and greatly appreciated.
- The paper is well written.

**Suggestions for minor improvements**:
- At the top of page 2, the authors appear to claim that Locatello et al. (2019b) states unsupervised disentanglement is not possible due to frequent violations of the assumption of independence of the source variables. However, from my understanding, the main argument of Locatello et al. (2019b) relates to the identifiability of ‘disentangled’ representations when no supervision is provided (even in the case of fully independent generative factors). I agree that correlated source variables further makes this identification more challenging, but the specific framing of this sentence appears incorrect. I would be happy to hear the authors comments on this if I am misunderstanding.
- It is somewhat disappointing to use an identity mixing matrix in the toy examples. Is there a reason why this analysis could not be done with randomly initialized matrices?

Overall, the paper makes interesting points in regards to standard mutual information minimization for disentanglement and gives strong evidence that attribute-conditional mutual information minimization may be a promising path forward for generalization under correlation shifts. I believe it would make a good addition to the workshop and thus recommend acceptance.

---

### Official Review · Reviewer_8tUU · 2022-03-15
**Disentanglement and generalization under correlation shifts**

**Rating:** 3
**Confidence:** 2

**Review:**

It is known that disentangling based on decorrelation or statistical independence cannot work when the true factors of variation are in fact correlated / dependent in some domain. This paper provides a very clear analysis of disentangling under correlation shifts using a simple linear Gaussian model. It is shown using this toy example that even when supervised data is available, robust disentanglement may fail, in the sense that when correlation patterns shift, performance degrades. Adding a mutual information minimization loss does not mitigate the issue and in fact can make it worse, but using conditional MI does resolve the issue. This idea is applied to the CelebA faces dataset (with artificially added correlations between attributes) using an adversarial approach for CMI minimization, and promising results are obtained.

---

### Decision · Program_Chairs · 2022-03-24

**Decision:**

Accept (Oral)

**Comment:**

Both reviewers agree about the high quality of the paper: the clarity of the presentation, the significance of the results, and the tight correlation to this workshop. Thus, this paper is given an oral presentation at the workshop. Congratulations!

The authors are encouraged to take the feedback by reviewer ZNS4 into account when preparing the camera-ready version of the paper.